# Separation of Anionic Chlorinated Dyes from Polluted Aqueous Streams Using Ionic Liquids and Their Subsequent Recycling

**DOI:** 10.3390/ijms241512235

**Published:** 2023-07-31

**Authors:** Barbora Kamenická, Petr Švec, Tomáš Weidlich

**Affiliations:** 1Chemical Technology Group, Institute of Environmental and Chemical Engineering, Faculty of Chemical Technology, University of Pardubice, Studentska 573, 532 10 Pardubice, Czech Republic; barbora.kamenicka@upce.cz; 2Department of General and Inorganic Chemistry, Faculty of Chemical Technology, University of Pardubice, Studentska 573, 532 10 Pardubice, Czech Republic; petr.svec2@upce.cz

**Keywords:** acid azo dye, Mordant Blue 9, Acid Yellow 17, ionic liquids, ion pairs, reduction, dehalogenation, hydrodechlorination, NaBH_4_, NiSO_4_, Raney Al-Ni alloy

## Abstract

The effect of ionic liquids on the separation of chlorinated anionic dyes such as Mordant Blue 9 (MB9) or Acid Yellow 17 (AY17) via ion exchange has been investigated in model aqueous solutions that simulate wastewater from the textile dyeing industry. The effect of ionic liquids chemical nature on the separation efficiency of mentioned dyes has been compared. It was found that especially ionic liquid based on quaternary ammonium salts comprising two or three long alkyl chains bound to the quaternary ammonium nitrogen (typically benzalkonium chloride or Aliquat 336) are very effective for the separation of both studied MB9 and AY17 from aqueous solution. In addition, the innovative technique has been developed for the reactivation of spent ionic liquids which is based on the chemical reduction of the formed ion pairs using NaBH_4_/NiSO_4_, NaBH_4_/Na_2_S_2_O_5_ or Raney Al-Ni alloy/NaOH. Thus, only NaBH_4_/NiSO_4_ in co-action with Al-Ni alloy enables both effective reduction of the azo bond and subsequent hydrodechlorination of emerging chlorinated aromatic amines. The efficiency of tested dyes separation or regeneration of ion pairs was evaluated by determination of the absorbance at wavelength of the maximum absorbance, of the Chemical Oxidation Demand (COD), and of the Adsorbables Organically bound Halogens (AOX). The formation of ion pairs or products of reduction and hydrodechlorination of these ion pairs has been studied using the ^1^H NMR and LC-MS techniques.

## 1. Introduction

Approximately over a hundred thousand types of dyes or pigments are estimated to be used within different dyeing processes [1]. Anionic or acid azo dyes (dyes bearing the N=N and -SO_3_^−^ groups) soluble in aqueous solutions are important synthetic colorants which are used for a wide spectrum of natural fibre fabric textile dyeing. Mordant dyes derived from acid dyes have been widely employed for the coloration of textile fibres, such as wool, silk, polyester, cotton, and some modified cellulose fibres [2]. The examples of mentioned mordant azo dyes are acid dye Mordant Blue 9 (MB9) and Acid Yellow 17 (AY17) depicted in Figure 1.

On average, the annual production of dyes exceeds 7.10^5^ tons [3]. However, in the manufacturing of the dyes or in the textile industry, typically about 5–10% of dyes are discharged with effluent and can potentially enter the environment [4]. This fact became an issue due to potential toxic effects of hardly biodegradable mordant dyes [5] and economic losses of dyes that remain in the wastewater [6].

Various methods have been developed to remove textile dyes from such wastewaters. The treatment processes based on degradation techniques comprise the oxidative [2,7,8] or reductive [2,9] conversions of dyes to less toxic and easily biodegradable products or even complete mineralization of dyes. Methods such as Fenton oxidation [7,8] or zero valent iron-based reduction [10,11] or aluminium/aluminium-valuable metal alloys [12,13] have been employed as well.

Several research groups [7,8,9] reported the high consumption of Fenton reagents for the effective decoloration (degradation) of selected (chlorinated) acid dyes. Compared to the chemical oxidation, the chemical reduction allows more selective reductive cleavage of the azo bonds [2,12]. In addition, the application of appropriate reductants serves for effective hydrodehalogenation of carbon-halogen bonds which are present in some acid azo dyes such as mentioned MB9 or AY17 [12,13].

As we reported previously [9,12,13], reductive hydrodehalogenation using aluminium alloys provides efficient dehalogenation of chlorinated aromatic compounds, such as MB9. The products of hydrodechlorination (HDC) can be degraded possibly in the biological stage of the wastewater treatment plant (WWTP) [14].

On the other hand, the separation processes offer the isolation of dyes from wastewater streams. Thus, the separation processes include adsorption [15], membrane processes [16] or innovative methods of extraction [17,18,19,20,21,22,23,24].

In recent years, the performance of available ionic liquids (ILs) have been extensively investigated within various applications (catalysts, synthesis of pharmaceuticals, biopolymer processing, batteries etc.) [25].

Some authors [17,18,19,20,21,22,23,24] reported the application of ILs in the extraction of acid dyes from aqueous solutions. Vijayaraghavan et al. [17], for the first time in 2006, reported on the effective extraction of selected acid azo dyes from leather industry wastewater using ILs. Since 2006, other authors [18,19,20,21,22,23,24] introduced efficient methods of acid dyes extraction using ILs. Different types of ILs were tested, such as imidazolium based ILs [18,19,20,22] or phosphonium based ILs [17,23,24]. The extraction efficiency can be also dependent on the structure of ILs and/or dyes [17,18,19,20,21,22,23,24]. However, most of the published works are associated with high doses of ILs necessary for the effective extraction of acid dyes [17,18,19,20,21,22,23].

In contrast with mentioned extraction methods, we verified earlier [8,25,26] that the innovative separation of anionic organic compounds from aqueous solutions using ion exchange mediated by ILs serves as ion exchanger.

It is based on the utilization of commercially available and inexpensive ILs, such as quaternary ammonium salts (R_4_NX), that are employed within the separation of acid dyes from model aqueous solutions simulating industrial textile wastewaters. The ion exchange of ILs with acid dyes provided less polar ion pairs (R_4_N.SO_3_-dye) which can be separated from aqueous solutions by sedimentation and/or filtration.

However, in recent years, emphasis has been placed on raw material recycling and the economic and ecological impacts of the processes of removal of pollutants from contaminated aqueous streams [6]. Due to the economy aspects of the extraction processes, Gharehbaghi et al. [19] reported on the possibility of ILs recovery using acidic stripping solutions and centrifugation leading to the separation of the aqueous solution of dyes and used ILs [20]. Thasneema et al. published that the heating of dye-ILs mixture above 40 °C leads to the precipitation of dyes. It offers the possibilities of reuse of ILs in a new cycle [23].

Nevertheless, we attempted to develop the innovative technique of ILs recycling applied in the above-mentioned ion exchange process. Our presumption was based on the reduction of separated ion pairs R_4_N.SO_3_-dye using appropriate reductive agents and subsequent recycling of formed ion pair species.

## 2. Results and Discussion

### 2.1. Separation of MB9 Using ILs Based on R_4_NXs

The preliminary experiments were focused on the separation of MB9 using R_4_NX (where R is the methyl group and/or -alkyl, -CH_2_Ph, -(CH_2_)_n_CH_3_ and X is Br or Cl). For comparison of structure dependence of these substances, the R_4_NX with different numbers of long alkyl chains bound to the quaternary ammonium nitrogen were scrutinized (structural drawings of R_4_NXs, Appendix A). These experiments were performed using model solutions of MB9 simulating industrial wastewater from the production of acid azo dyes [27,28].

The ion exchange between R_4_NX and MB9 anion is based on the formation of insoluble ion pairs (e.g., according to the production of ion pair (2) in Figure 1 below) that can be separated from aqueous solutions by simple filtration [8]. Formation of the respective ion pairs was proved by NMR spectroscopy of samples prepared from extracts of the ion pairs (for NMR spectra of MB9 and ion pairs (R_4_N)_2_.MB9 see Appendix A).

For example, the MB9 removal efficiency using Octyl_3_MeNCl was essentially quantitative. The other tested R_4_NXs provided lower removal efficiency than Octyl_3_MeNCl. The decoloration efficiency corresponded well to the removal efficiency of AOX, see Figure 2.

Compared with the above-tested R_4_NXs, the cheaper and commercially available ILs (structurally comparable with verified R_4_NX; see Appendix A) were employed within the separation of MB9 from model wastewater.

Selected commercial R_4_NXs comprising properties of ILs such as melting point below 100 °C are able to separate the MB9 dye from aqueous solutions. Especially, Aliquat 336 (A336, liquid at room temperature) or benzalkonium chloride (BzlkoniumCl, melting point 70–75 °C) offer interesting decoloration efficiency of MB9 solutions (see Figure 2). These ILs are not only comparable effective such as respective R_4_NX alternative, but they are also inexpensive and available for possibly practical applicability in a large scale. It was also observed that the effectiveness of decoloration depends on the structure of R_4_NXs itself. The decoloration efficiency of R_4_NX (on MB9 solutions) increases with the increasing number of long alkyl chains bound within the quaternary ammonium cation of the respective R_4_NXs (as demonstrated in Figure 2 above). Besides the above mentioned two ILs, commercial cationic surfactant Luviquat Mono LS (Luv. Mono LS) was recognized as effective ion-pairing agent for MB9 dye separation, too.

We supposed that the observed effective removal of MB9 by action of these 3 mentioned R_4_NXs is caused by the very limited solubility of the emerging ion pairs in water. Our assumption is further supported by the octan-1-ol/water partition coefficient (P_ow_) for ion pairs of R_4_N^+^ with the MB9 dye (see Figure 3). In almost all cases, much higher solubility of (R_4_N)_2_.MB9 in low polar octan-1-ol can be achieved using R_4_NX containing two and/or more long alkyl chains.

Even though the removal efficiency of MB9 increases with number of long alkyl chains within R_4_NX, the aqueous solutions of commercially available ILs with one or two long alkyl chains provide satisfactory separation capacity of MB9 from aqueous solutions. For example, commercially available aqueous solution of BzlkoniumCl provides a reasonable compromise between solubility in water and the separation efficiency of acid dyes from water (see Figure 2 and Figure 3).

Except the above mentioned R_4_NXs, we observed that even Luv. Mono LS is very effective for the removal of MB9 (see Figure 2). As can be seen from the Figure 3, the octan-1-ol/water partition coefficient for (Luv. Mono LS)_2_.MB9 is higher than octan-1-ol/water partition coefficient for structurally similar (AlkMe_3_N)_2_.MB9, but also it is lower than octan-1-ol/water partition coefficient for (A336)_2_.MB9 or (Bzlkonium)_2_.MB9. This surprising fact may be caused by the composition of commercial product Luv. Mono LS, which is supplied as the mixture of alkylated trimethylamine salts [29].

Efficient hydrophobic A336 is extremely viscous (viscosity 1500 mPa.s) and its accurate application is thus troublesome. The low solubility of the most effective R_4_NXs in water, along with their high viscosity, may cause problems with their application in praxis.

On the other hand, above-described Luv. Mono LS or BzlkoniumCl are not only commercially available, but are also supplied as aqueous solutions. These R_4_NXs (ILs) can be optionally used for the separation of MB9 from aqueous solutions without application of organic solvents [30] or more technologically demanding extraction [17,18,19,20,21,22,23,24].

In addition, the major advantage of Luv. Mono LS is its significantly lower price than that of Octyl_3_MeNCl (only 102 $/kg Luv. Mono LS [29] vs. 8600 $/kg Octyl_3_MeNCl [31]).

The described ion exchange-based separation technique requires much lower amounts of ILs compared with the conventional extraction methods described in the literature [16,17,18,20]. Appendix A compares such results mentioned in published articles dealing with extraction of acid dyes.

Following our research, we have found that the formation of ion pairs with 3,4-dichloraniline-6-sulphonic acid (DCA-6-SA—the reduction product of some acid azo dyes) with AlkBzlMe_2_NCl (ion pair 1) allows the >99% removal of AOX. Then, we compared the separation of MB9 using AlkBzlMe_2_NCl (ion pair 2) or AlkBzlMe_2_N salt of DCA-6-SA (independently prepared ion pair 2), as depicted in the Figure 1.

The removal efficiency of MB9 upon formation of (2) reaches 75.2%. On the other hand, using an independently prepared (2), the removal efficiency of MB9 decreased to 66.2%.

This comparison proves that ion pairs containing low weight anion enables ion exchange with sodium salt of the higher molecular weight anion (e.g., MB9). These findings introduce the possibility of recycling the R_4_NXs (ILs) from the separated ion pairs by reductive degradation of (R_4_N).dye ion pairs.

### 2.2. Reductive Hydrodechlorination of Ion Pairs and Recycling of IL-Based Species

Our recent observation that Al-Ni alloy provides reductive HDC of MB9 (according to the reaction in Figure 2) [9] could be important and useful for the sustainable recycling of separation agents based on the reduction and HDC of the formed ion pairs containing non-biodegradable halogenated anions such as in (R_4_N)_2_.MB9. The comparison of reductive/HDC methods for MB9 degradation are summarized in our previous work [9].

As we designed, the entire cycle of dye separation, regeneration and recycling of ion pair species consists of three steps.

The first step (i) comprises the above developed ion exchange between MB9 and virgin IL (R_4_NX). As we mentioned vide supra, upon the ion-exchange reaction of R_4_NX with MB9, ion pairs (R_4_N)_2_.MB9 are formed (along with an inorganic salt NaX).

The second step (ii) consist of the reductive degradation in methanol solution accompanied by the hydrodehalogenation of the emerging ion pair (R_4_N)_2_.MB9. The products of (R_4_N)_2_.MB9 ion pair reduction have a lower weight of the anion and are far better soluble in water. The high solubility of reduced ion pairs in water was proved by the determination of partition coefficients octan-1-ol/water system (see Figure 4). Corresponding products of the azo bond reduction and HDC of MB9 are depicted in Figure 2.

After the removal of methanol by simple distillation, the final step (iii) includes the anion exchange reaction described in Figure 1 above (the exchange of the anion of R_4_N^+^ formed by the reduction with MB9 dye whose molecular weight is higher that of the anion of reduced (R_4_N)_2_.MB9).

In addition, the waste products of HDC are easier biodegradable and can be readily degraded using standard biological treatment step in WWTP [12,13].

Our initial experiments tested the possible reductive recovery of (R_4_N)_2_.MB9 using solid NaBH_4_ alone. AlkBzlMe_2_NCl, Alk_2_Me_2_NCl and Octyl_3_MeNCl were employed within the attempted separation of MB9.

The decoloration of MB9 aqueous solutions suggests that regenerated ion pair species within reduction using 50 mol of NaBH_4_ per 1 mol of the appropriate ion pair reached satisfactory rate of decoloration to the second cycle of recycling of ion pair species in the case of all tested R_4_NX (see Figure 5).

As one could expect, the separation efficiency of MB9 using regenerated IL-based separation species decreased with each subsequent recycling cycle.

Doubling the dosage of the sodium borohydride, which is added to ion pair species solutions does not increase the reduction efficiency of ion pairs and subsequent ion exchange along with the decoloration efficiency of MB9 solutions in the 3rd and 4th cycle of recycling (see Figure 5).

However, based on available results, the reductive regeneration of quaternary ammonium salts of halogenated organic (sulphonic) acids seems to be an effective and promising method for recovery of applied ILs in processes of separation of acid azo dyes.

Despite the efficiency of the decoloration of MB9 solution, the AOX parameter during the recycling of ion pair species did not reach satisfactory values. The removal of AOX in the case of ion exchange of MB9 with all tested R_4_NCls corresponds to the decoloration efficiency and it is not lower than 68.2%. Unfortunately, after the reduction of the generated respective ion pairs using NaBH_4_, the removal of AOX in the MB9 separation process significantly decreased to 43.6% (see Appendix A).

We have found that NaBH_4_ is suitable for azo bond reduction within ion pairs as represented for example by (Octyl_3_MeN)_2_.MB9. It provides effective decoloration of the ion pair (Octyl_3_MeN)_2_.MB9 in the reduction processes from the 1st to the 4th regeneration cycle (see Figure 6). Efficient reduction of the azo bond (decoloration of the ion pair > 99%) enables above-described effective subsequent recycling of applied ILs (see Figure 5 above).

MB9 is, however, a chlorinated organic compound. Due to this reason, emphasis should also be placed on the dehalogenation of this dye.

The fact that NaBH_4_ is not an effective HDC agent has been already discovered in our previous work [9] describing the different reductive agents for MB9 degradation. This findings was supported by the reduction of (AlkMe_3_N)_2_.MB9 using 50 mmol of NaBH_4_ per 0.5 mmol of the respective ion pair (100 mol of NaBH_4_ per one mol of (AlkMe_3_N)_2_.MB9); see Table 1 below.

Due to excellent HDC activity of the Al-Ni alloy [9,13], the reductive degradation of ion pairs accompanied potentially by HDC upon the using Raney Al-Ni alloy/NaOH was scrutinized. The practical method of HDC using the Raney Al-Ni alloy is described in ref. [12,13] in detail.

Subsequently, we focused on the reductive HDC of the ion pair (A336)_2_.MB9 using the mentioned Al-Ni alloy. Based on the AOX results before and after HDC of the ion pair (A336)_2_.MB9, the Al-Ni alloy provide AOX removal efficiencies > 99% (see Figure 7). The efficiency of HDC activity of Raney Al-Ni was also proved by the dehalogenation of (AlkMe_3_N)_2_.MB9 using 100 mol Al in the Al-Ni alloy + 150 mol NaOH. The removal efficiency of AOX was 99.6%; see Table 1 below.

Despite the effective reduction and dechlorination of ion pairs, the degradation method of MB9 using Al-Ni alloy is relatively expensive [9]. Thus, the innovative processes decreasing the Raney Al-Ni alloy consumption were investigated.

The combination of NaBH_4_ with Na_2_S_2_O_5_ provides interesting results of AOX removal from MB9 solutions [9]. On the other hand, the HDC of MB9 does not proceed completely [9].

Afterwards, the reduction step using NaBH_4_ in co-action with an additive such as Na_2_S_2_O_5_ (generation of a reducing agent—dithionite, see Equations (1) and (2) below) [9,32] performed before the catalytic dehalogenation step was tested for the reduction and possible HDC of the ion pair (AlkMe_3_N)_2_.MB9 according to reaction in Appendix A.
S_2_O_5_^2−^ + H_2_O → 2 HSO_3_^−^(1)
BH_4_^−^ + 8 HSO_3_^−^ + H^+^ → 4 S_2_O_4_^2−^ + H_3_BO_3_ + 5 H_2_O(2)

The reduction of 1 mmol of the ion pair (AlkMe_3_N)_2_.MB9 was tested using 20 mmol of NaBH_4_ with the addition of 50 mmol of Na_2_S_2_O_5_. After 1 h of action, 14.5 mmol of NiSO_4_ was added to the reaction mixture. Then another 10 mmol of NaBH_4_ was added to ensure the reduction and HDC.

The addition of NaBH_4_ to the solution of ion pair (R_4_N)_2_.MB9 containing NiSO_4_ produced Ni^0^ according to the reaction described by Equation (3). Ni^0^ produced in situ could possibly act as HDC catalysts, too. With advantage, the pH for efficient HDC was adjusted to the alkaline area using a commercially available 12% aq. solution of NaBH_4_ in 14 M NaOH [33].

2 NaBH_4_ + NiSO_4_ + 6 H_2_O → Ni^0^ + 7 H_2_ + Na_2_SO_4_ + 2 H_3_BO_3_(3)

As the LC-MS spectra (see Appendix A) revealed, the reduction of mentioned ion pair using the NaBH_4_/Na_2_S_2_O_5_/Ni^0^ system allows the reductive degradation of (AlkMe3N)_2_.MB9, but it does not provide efficient HDC.

Moreover, we observed that not even subsequent reaction with Al-Ni alloy (5 mmol of Al in Al-Ni alloy per 1 mmol of ion pair) not provide the quantitative HDC after 20 h of action (for LC-MS see Appendix A).

The combination of NaBH_4_/Na_2_S_2_O_5_/NiSO_4_/Al-Ni is not an adequate HDC system, despite the effective reduction of the azo bond. It can be presumably caused by the small dosage of the HDC agent Al-Ni alloy.

Due to the increasing consumption of the Al-Ni alloy required for the complete HDC of the ion pairs (R_4_N)_2_.MB9, the investigation was not continued for economic reasons (i.e., application of several different agents in combination with the relatively expensive Al-Ni alloy).

Due to the mentioned issues, the innovative reduction of (R_4_N)_2_.MB9 was developed using cheap NaBH_4_/NiSO_4_ reducing system.

We found that 25 mmol of NaBH_4_ with 5 mmol of NiSO_4_ per 0.5 mmol of (Luv. Mono LS)_2_.MB9 is not able to completely dechlorinate the tested ion pair. LC-MS analyses (see Appendix A) demonstrated that the ion pair (Luv. Mono LS)_2_.MB9 was successfully reduced. However, we also detected chlorinated species in the reaction mixture after the reduction using NaBH_4_/NiSO_4_. It is in good agreement with results of reductive dechlorination of MB9 dye using NaBH_4_/NiSO_4_ system [9].

Finally, an innovative reducing system NaBH_4_/NiSO_4_ in combination with Al-Ni alloy according to the reaction in Figure 3 was applied for the reduction and subsequent HDC of the ion pair (R_4_N)_2_.MB9.

The addition of 25 mmol NaBH_4_ + 5 mmol of NiSO_4_ in combination with 10 mmol Al in the Raney Al-Ni alloy per 0.5 mmol of (Octyl_3_MeN)_2_.MB9 afforded the efficient reduction of the azo bond and even the dehalogenation of the ion pair. As LC-MS analyses detected that this reducing system ensured complete HDC (according to the reaction depicted in Figure 3) of the tested ion pairs (for LC-MS spectra, see Appendix A).

The combination of NaBH_4_/NiSO_4_/Al-Ni alloy is thus the optimal reductive/HDC method for ion pairs. The addition of cheaper NiSO_4_ to NaBH_4_ decreases the consumption of more expensive sodium borohydride. Moreover, the primary azo bond reduction using NaBH_4_/NiSO_4_ decreases subsequent consumption of the Al-Ni alloy (see Table 1).

As follows from the subsequent recycling of the regenerated ion pair species, it can be reused in next cycles for an efficient removal of MB9 from aqueous solutions; see Figure 8. In the 2nd cycle of the recycling, the separation of MB9 using regenerated ion pair species formed by the reduction allows almost 60% removal efficiency of the respective dye. Figure 9 depicted the efficiency of ion pair reduction.

As Figure 8 depicts, the use of cheap commercial Luv. Mono LS provides comparable results similarly as the expensive Octyl_3_MeNCl. In addition, aq. solution of Luv. Mono LS offers easier application.

All the above-described findings were also employed within the treatment of the other acid chlorinated azo dye Acid Yellow 17. Complete HDC of the ion pair (Luv. Mono LS)_2_.AY17 (according to reaction in Appendix A) using NaBH_4_/NiSO_4_ (application of 12% aq. solution of NaBH_4_ in 14 M NaOH) and Al-Ni alloy was also verified using LC-MS analyses (see Appendix A). The recycling of the ion pair species (Luv. Mono LS)_2_.AY17 was efficient (64.4%) to the 2nd cycle; see Figure 10.

In comparison, the reduction of (Luv. Mono LS)_2_.AY17 or (Octyl_3_MeN)_2_.AY17 using NaBH_4_/NiSO_4_ alone (50–70 mol of NaBH_4_ + 10 mol of NiSO_4_ per 1 mol of ion pair) does not lead to the complete HDC of the tested ion pair (for the LC-MS spectrum see Appendix A). Table 1 compares the overall efficiency systems used for the reduction and HDC of respective ion pairs. Appendix A summarizes the prices of the most promising methods of reductive recycling of IL-based separation species. The combination of inexpensive and available Luv. Mono LS and verified regeneration and recycling of IL-based products of reduction using NaBH_4_/NiSO_4_ + Al-Ni alloy system offers cheap and possible applicable method of separating of dyes from wastewaters in a large scale.

## 3. Materials and Methods

The dyes Mordant Blue 9 and Acid Yellow 17 were purchased from commercial sources in a defined purity higher than 50% for MB9 and 60% for AY17 (Merck, Prague, Czech Republic).

The ionic liquids—Aliquat 336 (A336) 98%+, 50% aq. benzalkonium chloride (BzlkoniumCl), 20% aq. poly(dialkyldimethylammonium) chloride (poly(Alk_2_Me_2_N)Cl) and 30% aq. Luviquat Mono LS (Luv. Mono LS) and solid R_4_NXs—cetyltrimethylammonium bromide (Alk_3_MeNBr), dilauryldimethylammonium chloride (Alk_2_Me_2_NCl), benzyldimethylhexadecylammonium chloride (AlkBzlMe_2_NCl) and trioctylmethylammonium chloride (Octyl_3_MeNCl) in purity 98%+ were purchased from Merck. Octan-1-ol, Raney Al-Ni alloy, NaBH_4_, Na_2_S_2_O_5_, NiSO_4_.7H_2_O, deuterated chloroform (CDCl_3_) and dimethyl sulfoxide (DMSO-d_6_) were also purchased from Merck in purity higher than 98%. Additional chemicals and solvents in p.a. quality were obtained from a local supplier (Lach-Ner Co., Neratovice, Czech Republic).

The comparative experiments were performed in 250 mL round-bottomed flasks equipped with magnetic stirring bars on Starfish equipment (Radleys Discovery Technologies, Saffron Walden, UK) installed on a magnetic stirrer Heidolph Heistandard for parallel reactions. A tube filled with granulated charcoal was fitted onto the neck of the flasks.

### 3.1. Chemical Analysis

^1^H NMR spectra were recorded in CDCl_3_ or DMSO-d6 solutions on a Bruker Avance 500 spectrometer (equipped with a Z-gradient 5 mm Prodigy^TM^ cryoprobe) at frequency of 500.14 MHz or on a Bruker UltraShield^TM^ 400 spectrometer at frequency of 400.13 MHz at 295 K. The solutions were obtained by dissolving of dried ion pairs (resulting from the work-up of reaction mixtures) in 1 mL of CDCl_3_ or DMSO-d_6_. The values of ^1^H chemical shifts were calibrated to the residual signals of CDCl_3_ (δ(^1^H) = 7.27 ppm) or DMSO-d_6_ (δ(^1^H) = 2.50 ppm). Mutual content of constituents in obtained mixtures was established using the integration of the corresponding area of resonances.

Products of the reduction/HDC of ion pairs were detected using the LC-MS technique. The water samples were properly diluted and 2 μL of diluted sample was injected in column Hypersil Gold C4 100 × 3 mm. The LC separation of detected species was performed on a LC-MSD TRAP XCT Plus system (Agilent Technologies, Santa Clara, CA, USA) and it was carried out with a gradient program using (A) 5 mM CH_3_COONH_4_, (B) 95% CH_3_COONH_4_ and (C) acetonitrile at a flow rate of 0.5 mL/min and at temperature of 40 °C. The detected species were completely separated under the established gradient program in 40 min. The UV detection was carried out at wavelength of 254 nm. The LC system was connected to an Agilent 6490 Triple Q MS with an electrospray interface (ESI) and was operated in both positive and negative ionization mode, using the above-described gradient program in both cases. The MS settings were as follows: the temperature of drying gas 350 °C; drying gas 10 L/min; ion source gas 50 psi and mass range 50–1200 Da. Nitrogen was used as the nebulizer gas, curtain gas and collision gas.

A Hach DR2800 VIS spectrophotometer was employed within the absorbance measurements using 1 cm glass cuvettes. The concentrations of MB9 or AY17 in aqueous solutions were determined by measuring at wavelength of 516 nm or 400 nm, respectively. The concentrations of MB9 or AY17 in CH_3_OH solutions were determined by measuring at wavelength of 505 nm or 415 nm, respectively. The mixtures of reduced MB9 and reduced (AlkMe_3_N)_2_.MB9 were determined with UV-VIS spectrophotometer Spectroquant Prove 300 (Merck) at maximum wavelength (317 nm) using a quartz cuvette. The AOX (Adsorbable Organically bound Halogens) were analysed according to the European ISO 9562 standard using Multi X 2500 analyser (Analytic Jena Co., Jena, Germany).

The decoloration efficiencies of MB9 or AY17 solutions were calculated according to the Equation (4):(4)$$DE=\left(1-\frac{A}{A_{0}}\right)\times 100$$
where *DE* is decoloration efficiency of the dye aqueous solution (%), *A* is measured absorbance after the removal process and *A*_0_ is initial absorbance of the dye solution. All the error bars in figures are calculated as the relative standard deviation (RSD) of decoloration efficiency. In all cases, the RSD were less than 6%.

The removal efficiency (%) of AOX from the model aqueous solutions of MB9 or AY17 was evaluated with respect to the Equation (5):(5)$$RE=\left(1-\frac{c}{c_{0}}\right)\times 100$$
where *RE* is removal efficiency of AOX (%), *c* is concentration of AOX in solution after the removal process (mg/L) and c_0_ is initial concentration of AOX in MB9/AY17 solution before removal process (mg/L).

### 3.2. Determination of Octan-1-ol/Water Partition Coefficients

An aqueous solution containing 1 mmol of MB9 was introduced to the round bottomed flask (in case of partition coefficient of ion pairs 2 mmol of R_4_NX per 1 mmol of -SO_3_^−^ group bound in the dye was added) and the total volume of aqueous phase was adjusted to 100 mL with demineralized water and the mixture was filled in using 100 mL of octan-1-ol. The prepared two-phase mixture was agitated at 400 rpm overnight and the immiscible phases were separated in a separatory funnel and the concentration of dye or ion pair was analysed using VIS spectroscopy. The partition coefficient (P_OW_) was calculated according to the Equation (6). Each experiment was performed three times; the presented values of log P_ow_ are calculated as mean value. The error bars in figures dealing with octan-1-ol/water partition coefficients presented the standard deviation of log P_ow_ value.
(6)$$logP_{OW}=\frac{c_{octanol}}{c_{water}}$$

### 3.3. Separation of Acid Dyes from Aqueous Solutions Using ILs

The aqueous solution of MB9 (120 mL of 4.2 mM stock solution) in demineralized water was stirred with 1.2 mmol of R_4_NX (ILs) in 50 mL methanol at 500 rpm at 25 °C for 4 h. After the ion exchange, methanol was distilled off. Then aqueous phase was filtered and its absorbance was measured at the wavelength of the maximum absorbance and/or AOX. In the case of the NMR analysis of ion pairs, the reaction mixtures after ion pair formation were extracted with one portion of CH_2_Cl_2_ (1 × 100 mL) for 20 h. Afterwards, the dichloromethane phase was evaporated to dryness.

### 3.4. Regeneration and Recycling of IL-Based Species

The ILs were dissolved in 50 mL of methanol and the aqueous solution of MB9 or AY17 was added to the methanolic IL solution. The dosage of R_4_NX was adjusted according to the number of -SO_3_^−^ groups bound in the dye structure. The reduction of ion pair in methanol solution was carried out using different reduction systems (for more details see Table 1). After the chemical reduction and/or HDC of ion pair, methanol was distilled off and ion pair species obtained by the reduction was used for the next decoloration step. The amounts of R_4_NX, dyes and reducing agents and time period of separation/reduction reactions is described in Section 3.

Reaction mixture was stirred at 500 rpm at 25 °C for the appropriate period of time, filtered and absorbance at the wavelength of the absorption maximum was determined in the filtrate and/or concentration of AOX was determined. The identity of products after the reduction and HDC was established using the LC-MS techniques.

## 4. Conclusions

This article has dealt with the separation of chlorinated acid azo dyes MB9 and AY17 from aqueous solutions using commercially available ILs (R_4_NXs) and with subsequent reductive treatment of separated ion pairs R_4_N.dye accompanied by the re-use of the IL-based reduction products.

Based on the results obtained, it can be concluded that:(i)1.2 mmol of R_4_NX is able to separate 0.5 mmol of MB9 after 4 h of action and efficiency of MB9 separation increases with increasing number of long alkyl chains bound within the R_4_NX species;(ii)commercially available and inexpensive ILs such as BzlkoniumCl or Luv. Mono LS (structurally also R_4_NX) provide the compromise between their removal efficiency for MB9 dye (82.1% and 90%, respectively) and solubility in water (log P_ow_ of the respective ion pair 0.91 and 0.67, respectively) required for the practical use;(iii)solid NaBH_4_ combined with NiSO_4_ (50 mol NaBH_4_ + 10 mol NiSO_4_ per 1 mol of (R_4_N)_2_.MB9) have been innovatively used to reduce the formed ion pairs;(iv)in the next step, the Raney Al-Ni alloy (20 mol Al in Al-Ni alloy per 1 mol of (R_4_N)_2_.MB9) can be used as an efficient HDC agent;(v)combination of the Raney Al-Ni alloy in co-action with NaBH_4_/NiSO_4_ decrease the consumption of Raney Al-Ni alloy while the price of the proposed reductive HDC system is over 45% lower compared to the application of high dosages of the Al-Ni alloy alone;(vi)recycling of the regenerated IL-based reduction products is effective up to the third cycle of its repeated application;(vii)spent Ni catalysts can be recycled [34] and re-applied as NiSO_4_ in co-action with NaBH_4_ while used methanol can be recycled by simple distillation; these steps potentially decrease the amount of produced waste and economic costs of raw materials in the potential large scale applications.

Generally, the proposed processes provide an effective method for the efficient removal of the halogenated aromatic contaminants such as acid dyes from wastewater. Recycled IL-based species obtained via the reduction/HDC of the respective ion pairs provides effective removal abilities for acid azo dyes separation and subsequent reductive and biological treatment.

## Data Availability

Not applicable.

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
