# Peer review of "Separation of Anionic Chlorinated Dyes from Polluted Aqueous Streams Using Ionic Liquids and Their Subsequent Recycling"

_ijms, 2023, doi:10.3390/ijms241512235_

Round 1

Reviewer 1 Report

In this work, authors have claimed to separate polluted water containing anionic dyes using ionic liquids. The authors claim this to be an ion exchange mechanism rather than a simple extraction. I have some serious reservations regarding the study as below.

1) Is benzalkonium chloride and aq. Poly (dialkyldimethylammonium) chloride are ILs? Same applied to many other 'ILs' used in the study. Please provide melting points for EACH!

2) How can ILs perform 'ion exchange' without solid resin support to hold on to either ion?  Upon dissolution into ILs, any ionic compound generates four possible ionic species with similar affinity, which is not the case for the 'extraction' mechanism.

Minor editing is required for better clarity. 

Author Response

The authors are grateful to the reviewer for their time, valuable comments and suggestions which helped to improve this manuscript.

Our responds to rewiever questions/comments:

In this work, authors have claimed to separate polluted water containing anionic dyes using ionic liquids. The authors claim this to be an ion exchange mechanism rather than a simple extraction. I have some serious reservations regarding the study as below.

Dear reviewer, the ion exchange process was documented by comparison of NMR spectra of starting ionic liquid and ion pair Dye-SO3NR4 produced by ion exchange between:

Dye-SO3Na + R4NCl → Dye-SO3NR4 + NaCl

The equilibrium of this reaction is completely shifted to the insoluble ion pair Dye-SO3NR4 in case, if R = long lipophilic alkyl chain.

In addition, you can observe differences in octan-1-ol/water partition coefficients (log Pow) comparing acid dye and prepared and separated ion pairs (see Figure 3).

1) Is benzalkonium chloride and aq. Poly (dialkyldimethylammonium) chloride are ILs? Same applied to many other 'ILs' used in the study. Please provide melting points for EACH!

 Benzalkonium chloride includes in room temperature ionic liquid classification (see for example the article: M.A. Deyab, Y.M. Moustafa, M.I. Nessim, Nesreen A. Fatthallah, Noha M. Asaad Bagato: New series of ionic liquids based on benzalkonium chloride derivatives: Synthesis, characterizations, and applications. Journal of Molecular Liquids 313 (2020) 113566.

There is quite problematic to find melting points of room temperature ionic liquids, in many cases, the melting point was not determined.

Poly (dialkyldimethylammonium) chloride is the polymerized IL (during polymerization, the formation of N,N-dimethylpyrrolidinium chloride skeleton is documented).

2) How can ILs perform 'ion exchange' without solid resin support to hold on to either ion?  Upon dissolution into ILs, any ionic compound generates four possible ionic species with similar affinity, which is not the case for the 'extraction' mechanism.

 You are right, the simple mixing of equimolar quantities of Dye-SO3Na with R4NCl produces the mixture of 2 cations and 2 anions.

However, if sparingly soluble ionic liquid R4NCl is used, the same process conducted in aqueous solution is accompanied by precipitation process of insoluble ion pair Dye-SO3NR4 (usually in the form of tarry like matter). The inorganic ions (Na+ and Cl-) remain in aq. solution. This is the principle of described anionic dyes separation.

Reviewer 2 Report

The effect of ionic liquids on the separation of chlorinated anionic dyes such as Mordant Blue 9 or Acid Yellow 17 by ion exchange in model aqueous solutions simulating wastewater from the dye industry was investigated. The topic is interesting and the problem is relevant. The research described in the manuscript is innovative, but questions arise. Why did the authors choose these ionic liquids? Can this be further explained in the text? My concern is the applicability of the proposed solution on a larger scale. Can the authors comment on this? Statistical analysis is missing. What are the errors in the results presented in the figures? In my opinion the conclusions should be rewritten - it's an abstract, not the conclusions of the study. There are a few errors in the text, for example Bzlkonium is changed to Bzkonium in the figures - careful editing of the paper is recommended.

Author Response

The authors are grateful to the reviewer for their time, valuable comments and suggestions which helped to improve this manuscript.

Our responds to rewiever questions/comments:

  • Why did the authors choose these ionic liquids? Can this be further explained in the text?

As we mentioned in article in Section 2.1, the preliminary experiments tested the separation of MB9 using R4NX (where R is the methyl group and/or -alkyl, -CH2Ph, -(CH2)nCH3 and X is Br or Cl). These R4NX were chosen due to different numbers of long alkyl chains bound to the quaternary ammonium nitrogen for testing the influence of numbers of long alkyl chains in R4NX structure on the separation efficiency. Compared to the verified R4NX, we focused on the application of ILs structurally similar to the mentioned R4NX, see Figure S1 in Supplementary Materials. The applied ILs such as Luv. Mono LS, A336, BzlkoniumCl are not only comparable effective such as respective R4NX, but they are also cheaper and available for possibly practical applicability in a large scale. For example, as we mentioned in article very effective Luv. Mono LS is significantly cheaper than that of Octyl3MeNCl (only 102 $/kg Luv. Mono LS [29] vs. 8600 $/kg Octyl3MeNCl [31]). Because of these reasons, mentioned R4NX and ILs (structurally also defined by R4NX formula) were chosen for this study. It was revised in an article in Section 2.1.

  • My concern is the applicability of the proposed solution on a larger scale. Can the authors comment on this?

In this article, proposed method was verified in the cooperation with local company producing of chlorinated azo dye. Thus, this innovative technique was also tested for possibly practically application; see our patent [26]. The combination of cheap and simply available Luv. Mono LS and described regeneration and recycling of ion pairs R4N.dye using NaBH4/NiSO4 + Al-Ni alloy system offers cheaper and possibly applicable method of separating of dyes from wastewaters in a large scale. However, for these cases the additionally study should be optionally required. It was revised in an article in Section 2.1 and 2.2.

  • Statistical analysis is missing. What are the errors in the results presented in the figures?

Statistical analysis was added, see Figures 2, 3, 4, 8, 9. The error bars in figures dealing with octan-1-ol/water partition coefficients are calculated as standard deviation of log Pow value. The error bars of decoloration efficiency presented the relative standard deviation of DE % (in all experiments RSD < 6 %).  Nevertheless, due to relatively small deviation of experiments, time consuming analysis of AOX or in the preliminary experiments (e.g. Figures 5,6,7), deeper statistical analysis could not be required. The errors in the results can be caused by performing of experiments, for example stirring, dosage, analysis, etc.

  • In my opinion the conclusions should be rewritten - it's an abstract, not the conclusions of the study.

It was revised in an article; see Conclusions.

  • There are a few errors in the text, for example Bzlkonium is changed to Bzkonium in the figures - careful editing of the paper is recommended.

It was revised in an article.

Round 2

Reviewer 1 Report

1) Benzalkonium chloride is a crystaline solid with a melting point of 160. https://www.scbt.com/p/benzethonium-chloride-121-54-0

The authors must have used an aqueous solution, and hence it is in liquid form. The ref. 2 M.A. Deyab et al./Journal of Molecular Liquids 313 (2020) 113566 is for "benzalkonium chloride-based ILs". 

Let me tell you the fundamentals of ILs. Any ionic compound with Cl as an anion can't be a RTIL. The lowest MP of Cl-based IL is [Bmim]Cl (around 70C).

2) The easy way to determine the melting point of any compound is simple. Freeze-dry or rotavap overnight and then conduct DSC.

3) What does the author mean by "sparingly soluble IL"? ILs 

4) What is the meaning of 'The reduction of ion pairs in methanol solution was carried out using different reduction systems'? (L552)

5) Next time, please provide a clean version of MS (without track changes and comments).

Author Response

  • Benzalkonium chloride is a crystaline solid with a melting point of 160. https://www.scbt.com/p/benzethonium-chloride-121-54-0

Dear reviewer, we are very sorry, however, this seems to be misunderstanding. We experimentally proved the typical IL benzalkonium chloride (N-alkyl-N-benzyl-N,N-dimethylammonium chloride, alkyl = C12-C14. CAS: CAS:85409-22-9). This mixture of chemical compounds melts around 70oC. See for example: Benzalkonium chloride CAS#: 85409-22-9 (chemicalbook.com) (M.p.: 73 oC, Bzlkonium chloride)

Mentioned benzethonium chloride is quite different compound (N-Benzyl-N,N-dimethyl-2-{2-[4-(2,4,4-trimethylpentan-2-yl)phenoxy]ethoxy}ethan-1-aminium chloride, CAS: 121-54-0), with melting point 160.

The authors must have used an aqueous solution, and hence it is in liquid form. The ref. 2 M.A. Deyab et al./Journal of Molecular Liquids 313 (2020) 113566 is for "benzalkonium chloride-based ILs". 

You are right, we used much more suitable 50% aq. solution due to its simple miscibility and solubility with treated aq. solutions (waxy anhydrous Bzlkonium chloride is covered with insoluble layer of ion pair immediately after addition to the aq. solution of anionic dye).

Let me tell you the fundamentals of ILs. Any ionic compound with Cl as an anion can't be a RTIL. The lowest MP of Cl-based IL is [Bmim]Cl (around 70C).

The commercial Aliquat 336 or benzalkonium chloride are industrial products defined as alkylated quaternary methylammonium chlorides (see for example Sigma-Aldrich catalogue:  https://www.sigmaaldrich.com/CZ/en. 

It is well known that mixtures of compounds pose lower achieve lower melting point compared with pure chemicals. For example, A336 (Aliquat 336, methyltrialkylammonium chloride) is the viscous liquid at room temperature.

  • The easy way to determine the melting point of any compound is simple. Freeze-dry or rotavap overnight and then conduct DSC.

We agree, you are right.

  • What does the author mean by "sparingly soluble IL"? ILs 

The tested ILs provide the limited solubility in water. For example, trialkylmethylammonium salts (Aliquat 336) provide only very limited solubility in water; their solubility in water is below 0.005 mol L-1 (see for example the article: Mikkola, J. P., Virtanen, P., & Sjöholm, R. (2006). Aliquat 336®—a versatile and affordable cation source for an entirely new family of hydrophobic ionic liquids. Green Chemistry, 8(3), 250-255.). It was revised in our article.

4) What is the meaning of 'The reduction of ion pairs in methanol solution was carried out using different reduction systems'? (L552)

 The different reduction systems such as NaBH4 alone, NaBH4 + Na2S2O5, NaBH4 + NiSO4, Al-Ni alloy etc were compared. Due to low solubility of formed ion pairs in water, the reduction has to carried out in methanol. It was specified in an article.

5) Next time, please provide a clean version of MS (without track changes and comments).

Sorry for the effected inconvenience. This new version is prepared according to your recommendations.

Round 3

Reviewer 1 Report

the revised version is now suitable for publication in its current form.